# Reduced matrix rigidity promotes neonatal cardiomyocyte dedifferentiation, proliferation and clonal expansion

Yfat Yahalom-Ronen[1,2], Dana Rajchman[1], Rachel Sarig[1], Benjamin Geiger[2]*, Eldad Tzahor[1]*

[1]Department of Biological Regulation, Weizmann Institute of Science, Rehovot, Israel; [2]Department of Molecular Cell Biology, Weizmann Institute of Science, Rehovot, Israel

**Abstract** Cardiomyocyte (CM) maturation in mammals is accompanied by a sharp decline in their proliferative and regenerative potential shortly after birth. In this study, we explored the role of the mechanical properties of the underlying matrix in the regulation of CM maturation. We show that rat and mouse neonatal CMs cultured on rigid surfaces exhibited increased myofibrillar organization, spread morphology, and reduced cell cycle activity. In contrast, compliant elastic matrices induced features of CM dedifferentiation, including a disorganized sarcomere network, rounding, and conspicuous cell-cycle re-entry. The rigid matrix facilitated nuclear division (karyokinesis) leading to binucleation, while compliant matrices promoted CM mitotic rounding and cell division (cytokinesis), associated with loss of differentiation markers. Moreover, the compliant matrix potentiated clonal expansion of CMs that involves multiple cell divisions. Thus, the compliant microenvironment facilitates CM dedifferentiation and proliferation via its effect on the organization of the myoskeleton. Our findings may be exploited to design new cardiac regenerative approaches.

*For correspondence: benny. geiger@weizmann.ac.il (BG); eldad.tzahor@weizmann.ac.il (ET)

**Competing interests:** The authors declare that no competing interests exist.

**Reviewing editor**: Deepak Srivastava, Gladstone Institute of Cardiovascular Disease, United States

## Introduction

During the early postnatal period, cardiomyocytes (CMs) undergo a switch from a proliferative, hyperplastic mode to non-proliferative, hypertrophic growth that persists throughout life (*Li et al., 1996*; *Soonpaa et al., 1996*; *Soonpaa and Field, 1998*). This process is accompanied by multiple, highly synchronized cellular changes. The expression of cell-cycle and embryonic markers falls precipitously in CMs (*Walsh et al., 2010*), while the expression of CM differentiation genes increases, along with the appearance of sarcomeres, contractile units that undergo cross-striation to form a functional myoskeletal system within CMs. Further, mechanical and electrical cell-to-cell communication between CMs is established via intercalated discs, containing gap junctions, adherens junctions and desmosomes (*Noorman et al., 2009*; *Sheikh et al., 2009*). In parallel, up-regulation of extracellular matrix (ECM) components, as well as increased cross-linking of matrix proteins, also occurs postnatally, leading to an overall stiffening of the heart tissue (*Janmey and Miller, 2011*; *Swift et al., 2013*; *Majkut et al., 2014*). The mechanisms whereby these complex developmental processes are regulated and coordinated, and their putative effects on CM maturation, are yet enigmatic.

Over the first week of postnatal life in mice, most CMs exit the cell cycle and differentiate (*Soonpaa et al., 1996*), although some CMs undergo an additional burst of proliferation in the pre-adolescent period, driven by the thyroid hormone (*Naqvi et al., 2014*). Mammalian CMs display distinct cycling phases: a fetal/neonatal phase in which nuclear division (karyokinesis) is immediately followed by cell division (cytokinesis), and a later neonatal phase, in which karyokinesis proceeds without cytokinesis, leading to binucleation (*Soonpaa et al., 1996*; *Li et al., 1997a*, *1997b*; *Zebrowski and Engel, 2013*). Binucleation, hypertrophic growth, increased myofibrillar organization, and cell cycle withdrawal are

**eLife digest** Heart muscle contracts and relaxes in a regular rhythm to pump blood around the body. Soon after birth, the cells that form our heart muscle stop multiplying. As we grow, these cells increase in size and their internal skeleton—called myoskeleton—becomes more complex, to withstand the demands of pumping more blood. However, because the cells can no longer divide, the body is unable to replace heart muscle cells that are damaged by a heart attack or other illness. This lack of ability to regenerate heart muscle is a major challenge for medicine. While researchers have documented many of the changes that occur in heart muscle cells (known as cardiomyocytes) after birth, it is not known exactly what triggers these changes.

A network of proteins and other molecules—also known as a matrix—surrounds the cardiomyocytes and affects their behavior. Here, Yahalom-Ronen et al. investigated the degree to which the mechanical properties of this matrix affect the ability of cardiomyocytes to divide. In the experiments, cardiomyocytes from newborn rodents were grown on matrices with different rigidities. The cells grown on rigid matrices stopped dividing and became larger with a more robust myoskeleton. These cells also contained two nuclei, which indicates that these cells have become mature cardiomyoctyes.

In contrast, heart cells grown on a softer matrix continued to multiply. These cells also began to lose some of the features that distinguish mature cardiomyocytes from the cardiomyocytes found in embryos. Next, Yahalom-Ronen et al. treated the cardiomyoctes with a drug that stops them from contracting, which led to increases in cell multiplication.

Yahalom-Ronen et al.'s findings suggest that the stiffness of the matrix that surrounds heart muscle cells regulates their ability to divide and mature. In the future, these findings may pave the way towards the development of soft scaffolds that can stimulate the regeneration of adult human heart.

all manifestations of the differentiated state of adult CMs (*Ahuja et al., 2007*; *Naqvi et al., 2009*; *Zebrowski and Engel, 2013*), while the 'terminal differentiation' state is yet uncertain.

Naturally, there are physiological advantages in the postnatal maturation of CMs. Nonetheless, loss of the proliferative potential of differentiated CMs creates a major barrier to cardiac regeneration after injury. In humans, myocardial infarction (MI) is a life-threatening disease leading to permanent loss of hundreds of millions of CMs, followed by an inflammatory response and formation of scar tissue, that progressively lead to cardiac dysfunction and heart failure (*Virag and Murry, 2003*; *Ausoni and Sartore, 2009*).

Cardiac regeneration does exist in lower vertebrates such as newts and fish. The zebrafish heart, for example, is able to fully regenerate after injury, without scarring (*Poss et al., 2002*). The lack of regenerative potential of the mammalian heart was challenged by Porrello and colleagues (*Porrello et al., 2011*), who showed that the neonatal murine heart displays a transient regenerative phase that diminishes within the first week after birth. Both mammalian and zebrafish heart regeneration is characterized by increased CM proliferation associated with sarcomere disassembly, attributed to a CM dedifferentiation process (*Poss, 2007*; *Jopling et al., 2010*, *2011*; *Porrello et al., 2011*).

Recent studies with diverse cell types indicate that the rigidity of the underlying (or surrounding) matrix strongly influences cell structure, cytoskeletal organization, migration, polarization, and regulation of gene expression and cell fate (*Pelham and Wang, 1997*; *Discher et al., 2005*; *Engler et al., 2006*; *Vogel and Sheetz, 2006*; *Engler et al., 2008*; *Geiger et al., 2009*; *Prager-Khoutorsky et al., 2011*; *Shin et al., 2011*). In particular, it was demonstrated that interaction with rigid surfaces (hundreds of kPa and stiffer) promotes the formation of large, integrin-mediated adhesions, and consequently, development of a well-organized and polarized cytoskeleton (*Prager-Khoutorsky et al., 2011*). Following this logic, we tested here the hypothesis that the decline in CM proliferative and regenerative capacities can be attributed to changes in the mechanical properties of the pericellular environment, which facilitate the assembly of a tightly organized myoskeletal structure. We hypothesize that matrix stiffening, which occurs in the heart after birth, and even more so following MI, is part of a mechanism that blocks CM cell cycle activity.

In this study, we discovered that soft matrices promote CM dedifferentiation, as evidenced by CM rounding, myofibrillar disassembly, increased CM cell division, and clonal expansion. In contrast, a rigid matrix facilitates CM differentiation, characterized by tightly packed arrays of long and aligned sarcomeric bundles, cell cycle arrest, and binucleation. Furthermore, we demonstrated that matrix rigidity specifically affects CM cytokinesis, but not karyokinesis. We next demonstrated that disruption of the highly organized architecture of the CM myoskeleton, using the myosin-II inhibitor blebbistatin, induced CM cell cycle re-entry. We suggest that the mechanical properties of the postnatal heart play a key role in the acquisition of the fully differentiated phenotype, and inhibition of the specialized contractile system in CMs could be used to promote CM dedifferentiation.

## Results

### Compliant substrates promote cardiomyocyte morphological changes and proliferation

To study the effect of matrix rigidity on CM cellular characteristics, we plated newborn (P1) rat CMs on PDMS substrates with different rigidities, ranging from stiff (2 MPa) to compliant (20 kPa and 5 kPa), and analyzed the sarcomeric and cellular organization. Myosin heavy chain (MHC), Myomesin (an M-band protein) and cardiac Troponin T (cTnT) were expressed in CMs cultured on both rigid and compliant substrates; however, CMs on the rigid substrate displayed aligned sarcomeres with defined and registered striations, whereas CMs on the compliant substrate had disorganized sarcomeres, with misaligned myoskeletal structures (*Figure 1A*). Furthermore, CMs cultured on the rigid substrate appeared larger, elongated, and mostly triangular, whereas CMs on the compliant substrate were round, smaller, and less polarized. CMs cultured on the 20 kPa substrate had intermediate parameters in terms of their polarization and spreading morphologies, compared to the 2 MPa and the 5 kPa substrates (*Figure 1B–D*).

To determine the effects of substrate rigidity on CM proliferation, we stained CMs with the cell-cycle markers Ki67 and phospho-histone-3 (PH3), together with cTnT and MHC (*Figure 1E,F*). Proliferating CMs were observed on all tested matrices (Ki67+/cTnT+, and PH3+/MHC+) (*Figure 1E,F*, respectively). CMs cultured on either glass or on the stiff (2 MPa) substrate presented similar proliferative capacities (data not shown). However, CM proliferation was increased on the compliant substrates (5 kPa and 20 kPa) by ~50–65%, respectively, relative to the rigid substrate (*Figure 1G*, visualized with Ki67), and by ~25–110% on the 5 kPa and 20 kPa substrates, respectively, marked by PH3+ MHC+ CMs (*Figure 1H*). Taken together, these findings suggest that myoskeletal disassembly, a tendency toward cell rounding, and increased CM proliferation are more compatible with compliant matrices, in comparison with the rigid matrix.

Inspired by the important finding of the transient regenerative potential in the neonatal mouse heart (*Porrello et al., 2011*), we established a mouse culture system of P1 neonatal CMs cultured on 2 MPa, 20 kPa, and 5 kPa substrates. As in rat CMs, mouse neonatal CMs grew on all substrates, and developed normal beating within 48–72 hr after seeding. We quantified the percentage of proliferating Ki67+ cTnT+ CMs, and observed that both the 5 kPa and 20 kPa substrates facilitated CM proliferation by ~30–50%, respectively, relative to the rigid 2 MPa substrate (*Figure 1I,J*). There was no significant difference in the area of cells plated on the different substrates (*Figure 1—figure supplement 1A*). Similar to the rat system, CMs cultured on the rigid substrate were elongated and more polarized, compared to those on the compliant substrates (*Figure 1—figure supplement 1B*). A decrease in CM cell perimeter, indicative of roundness, was observed on the compliant substrates, compared to the rigid one (*Figure 1—figure supplement 1C*). Hence, compliant substrates promote CM proliferation as well as alterations in CM cell shape, in both rat and mouse neonatal cultures.

### Distinct rigidity-dependent cardiomyocyte proliferation mechanisms

Various proliferation markers (e.g., Ki67 shown in *Figure 2A*, BrdU, PH3, and Aurora B), are widely used for cell-cycle assessment and quantification of cell proliferation; however, they sometimes prove insufficient, especially in CMs in which polyploidization and binucleation are natural outcomes of cell proliferation in the postnatal period (*Bersell et al., 2009*; *Zebrowski and Engel, 2013*). In order to explore the effects of matrix rigidity on CM cell division and the formation of new CMs, we established a live-cell imaging system of CMs derived from transgenic mice expressing the R26R-tdTomato reporter under the regulation of the *Myh6* promoter (*Myh6-Cre;R26R-tdTomato-lox*) (*Figure 2B*).

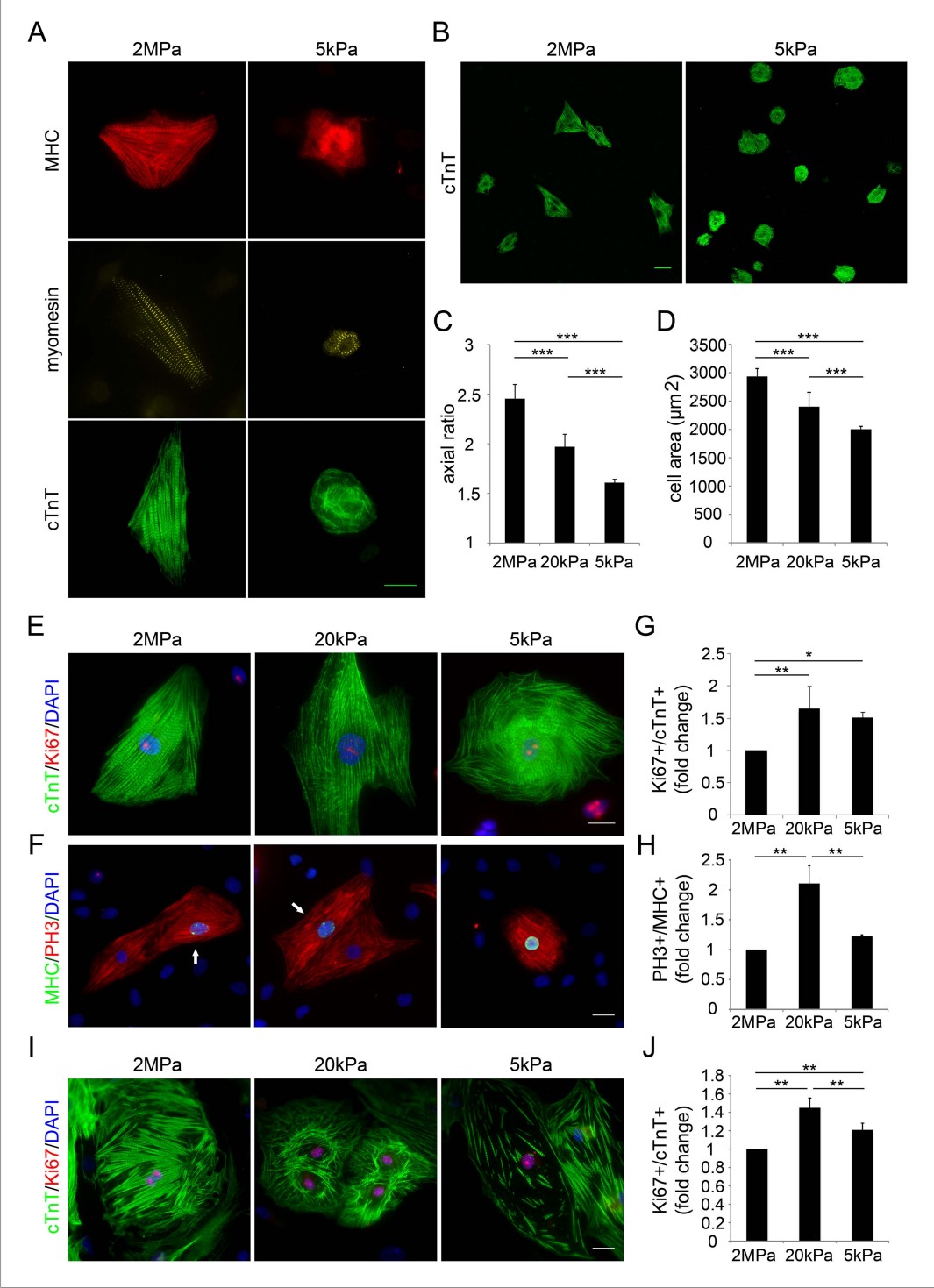

**Figure 1**. Compliant substrates alter neonatal cardiomyocyte cell shape and sarcomeric organization, and promote their proliferation. (**A**) Representative images of the sarcomeric patterns of MHC (top), myomesin (middle), and cTnT (bottom) on rigid 2 MPa (left) and soft 5 kPa (right) substrates. Scale bar: 20 μm. (**B**) An overview of neonatal P1 rat CMs on rigid (left) and soft substrates (right). Scale bar: 50 μm. (**C**) Axial ratio of P1 rat CMs (**D**) Cell area of P1 rat CMs (n = 1,108, for **C**, **D**). (**E**, **F**) Representative immunofluorescence images of proliferating P1 rat CMs on various PDMS substrates. Scale bar: 20 μm. CMs were stained for (**E**) sarcomeric cTnT and proliferation marker Ki67, or (**F**) cardiac MHC and PH3 (PH3+/MHC+ are denoted with arrows). (**G**) Quantification of Ki67+/cTnT+ P1 rat CMs on different rigidities (n = 4,165). (**H**) Quantification of PH3+/MHC+ P1 rat CMs on different rigidities (n = 1,964)
*Figure 1. continued on next page*

*Figure 1. Continued*

(**I**) Representative images of neonatal P1 mouse CMs on 2 MPa, 20 kPa, and 5 kPa substrates. (**J**) Quantification of Ki67+/cTnT+ P1 mouse CMs on different rigidities (n = 11,876). Statistical significance was determined using ANOVA followed by post-hoc Tukey's (HSD) test. Results are marked with one asterisk (*) if $p < 0.05$, two (**) if $p < 0.01$, and three (***) if $p < 0.001$.

The following figure supplement is available for figure 1:

**Figure supplement 1**. Compliant substrates alter P1 mouse cardiomyocyte cell shape, polarity, and sarcomeric organization.

Live-cell video microscopy revealed two distinct CM cell-cycle phenotypes: karyokinesis followed by binucleation (*Figure 2C*, *Video 1*), as opposed to karyokinesis followed by cytokinesis, resulting in the formation of two new CMs (*Figure 2D*, *Video 2*). In the first, CMs were relatively large, spread, and immotile (*Figure 2C*: 50′, 60′). These CMs underwent karyokinesis, leading to the formation of binucleated CMs (*Figure 2C*: 50′, 60′), while remaining well spread and attached to the substrate (*Figure 2C*).

In contrast, CMs that completed cell division (cytokinesis) underwent a step of mitotic rounding (*Figure 2D*), which is common for most proliferating cells (*Lancaster and Baum, 2014*); moreover, these CMs often underwent consecutive cell divisions. Strikingly, we found that compliant matrices did not affect nuclear cell division (karyokinesis), yet promoted cytokinesis and inhibited CM binucleation prominence, as determined by quantification of the division frequency, observed by live-cell imaging (*Figure 2E–G*).

In order to demonstrate an actual increase in CM number, we quantified the amount of CMs at the beginning and after 48 hr. A significant increase in the number of newly formed CMs was observed on the 20 kPa substrate, relative to the rigid 2 MPa (*Figure 2H*). Interestingly, we could observe rare events of cytokinesis even in binucleated CMs cultured on the 20 kPa substrate, resulting in two daughter CMs (*Figure 2—figure supplement 1E*). These successful events were also accompanied by mitotic rounding (*Figure 2—figure supplement 1*). Taken together, our findings demonstrate that culturing CMs on compliant matrices facilitate CM cell rounding and division (cytokinesis) that lead to formation of new CMs. In contrast, our results demonstrated that the rigid matrix promotes karyokinesis without cytokinesis, leading to CM binucleation (*Figure 2G*).

## Compliant matrices promote cardiomyocyte dedifferentiation

To further investigate the molecular status of CMs undergoing cytokinesis, we designed an assay that enabled us to correlate between the live imaging videos, in which we could visualize cell division processes (*Figure 2*), with molecular and lineage analyses of the dividing cells (*Figure 3*). Accordingly, correlated live cell-immunofluorescence microscopy was performed on CMs derived from P1 transgenic *Myh6-Cre;R26R-tdTomato-lox* mice cultured on 2 MPa, 20 kPa and 5 kPa substrates in grid-containing plates. The 'Tomato' cells represent CMs that express, or previously expressed, the Myh6 gene, a signature of CM differentiation. Under regular conditions, we detected almost 100% of dTomato+/cTnT+ double positive CMs (*Figure 3—figure supplement 1*).

Time-lapse imaging was performed for 48 hr, and immediately thereafter, cultures were fixed and immunostained for cTnT and Ki67. We first examined the time-lapse videos for CMs undergoing complete cell division (karyokinesis plus cytokinesis), and identified the two daughter cells by using the grid coordinates (*Video 3–5*). By correlating the last frame of each time-lapse video (*Figure 3A–C*) with cTnT staining, we revealed that the dividing CMs (tdTomato positive; *Figure 3A′,B′,C′*) lost cTnT expression, either completely (*Figure 3A″,B″*), or partially (*Figure 3C″*). This result is consistent with a CM dedifferentiation process, in which CMs, originating from the Myh6 lineage, lost cTnT expression (*Figure 3A″,B″,C″*). Furthermore, the majority of these cells expressed Ki67 on all three matrices, indicating that these CMs maintain a proliferative potential (*Figure 3A‴,B‴,C‴*).

To quantify CM dedifferentiation on the different matrix rigidities, we performed immunofluorescence analysis of P1 CMs cultured on the different substrates. We counted the number of tdTomato+/cTnT− CMs (*Figure 3D,E*) or MHC− (*Figure 3F,G*). A ~twofold increase in CM dedifferentiation

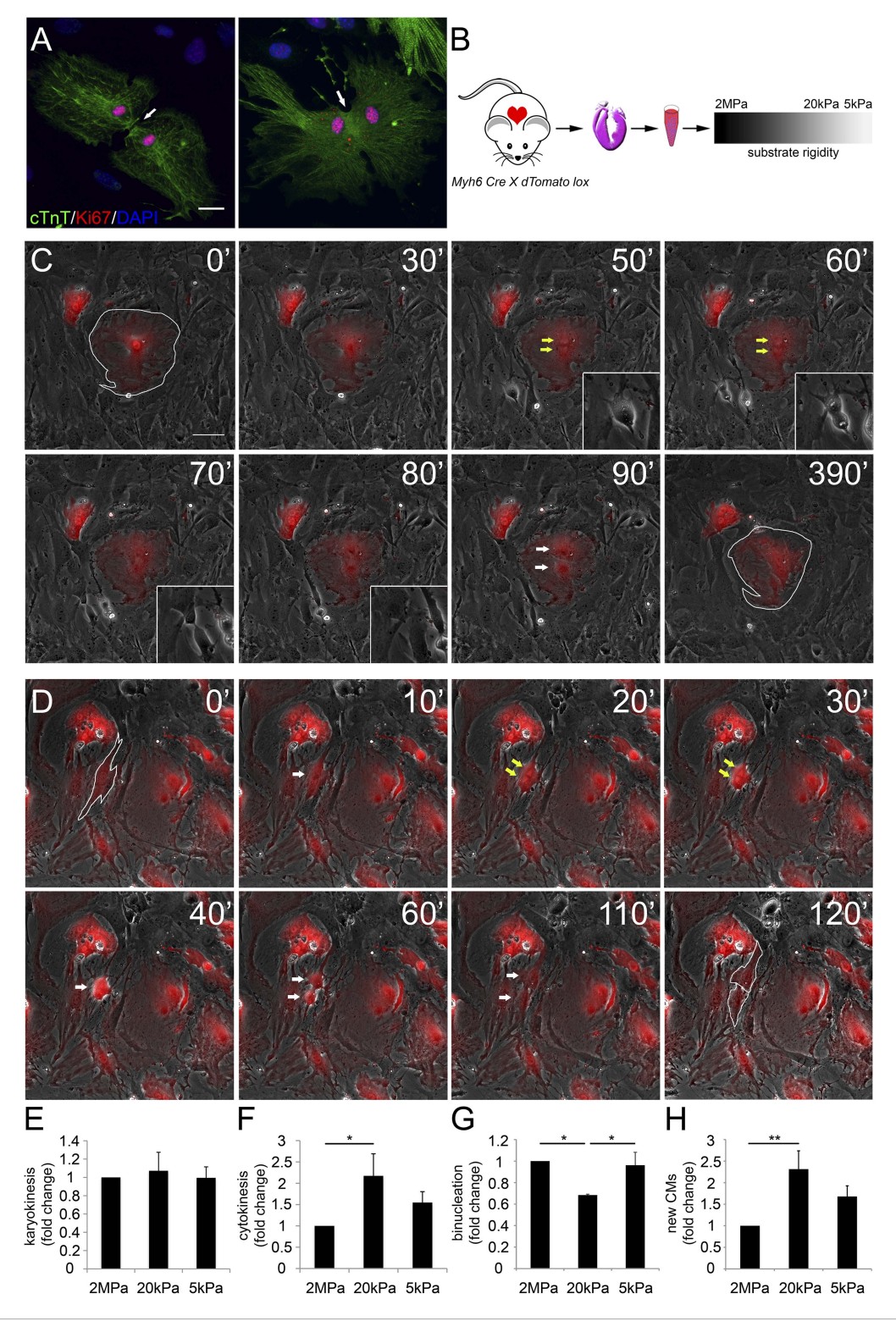

**Figure 2**. Distinct rigidity-dependent cardiomyocyte division mechanisms. (**A**) Immunofluorescent images of dividing neonatal CMs in culture. The newly formed CMs are still connected, and both nuclei are Ki67+ (red). (**B**) A schematic drawing of the experimental design. (**C**) Video frames of a P1 *Myh6-Cre;R26R-dTomato-lox* CM undergoing karyokinesis followed by binucleation, without changing its morphology. (Inset) A fibroblast undergoing typical cell division. (**D**) Video frames of a P1 *Myh6-Cre;R26R-dTomato-lox* CM undergoing karyokinesis, followed by

*Figure 2. continued on next page*

*Figure 2. Continued*

complete cytokinesis. The original CM is highlighted in 0′. The two new CMs are highlighted in 120′. Yellow arrows mark karyokinesis (20′, 30′); white arrows mark the new nuclei (40′), or new CM (60′,110′). (**E**) P1 mouse CM karyokinesis on different rigidities. (**F**) P1 mouse CM cytokinesis on different rigidities. (**G**) P1 mouse CM binucleation on different rigidities (n = 2,167 for **E, F, G**). (**H**) Quantification of new CMs on different rigidities (n = 2,878). Statistical significance was determined using ANOVA followed by post-hoc Tukey's (HSD) test. Results are marked with one asterisk (*) if $p < 0.05$, and two (**) if $p < 0.01$.
The following figure supplement is available for figure 2:

**Figure supplement 1**. Binucleated cardiomyocytes re-enter the cell cycle.

(Tomato+/cTnT− and Tomato+/MHC−) was observed on the 20 kPa, and an ∼1.5-fold increase, on the 5 kPa matrices, relative to the rigid matrix (*Figure 3E,G*). Moreover, all Myh6 lineage-positive (tdTomato+) CMs in the culture expressed Nkx2.5, indicating that these cells derived from the cardiac lineage (*Figure 3H*). Our findings, showing dividing CMs that downregulated cTnT and/or MHC sarcomeric proteins and express Nkx2.5, strongly suggest that the compliant matrices promote the dedifferentiation of CMs, consistent with their role in promoting cytokinesis (*Figure 3I*).

## Relaxation of the acto-myosin system induces cardiomyocyte proliferation

Our findings thus far suggest that disruption of the CM myoskeleton promotes CM cell cycle re-entry. We therefore tested whether inhibition of the CM myoskeleton function could boost CM proliferation. For that purpose, we used blebbistatin, a small molecule myosin II inhibitor that effectively blocks the actin-myosin interaction in several striated muscle (including CMs), in smooth muscle, and in non-muscle cells (*Dou et al., 2007*). Neonatal mouse CMs were cultured on PDMS substrates with the different rigidities, left to adhere for 72 hr, and then treated with 20 μM blebbistatin for 24 hr. After 24 hr of treatment, we fixed the cells and immunostained them for cTnT and Ki67. We found that blebbistatin severely disrupted CM myoskeleton morphology, leading to massive changes in CM cell shape, manifested by major flattening, dramatic fragmentation of the sarcomeric arrays, and complete loss of striations (*Figure 4A,B*).

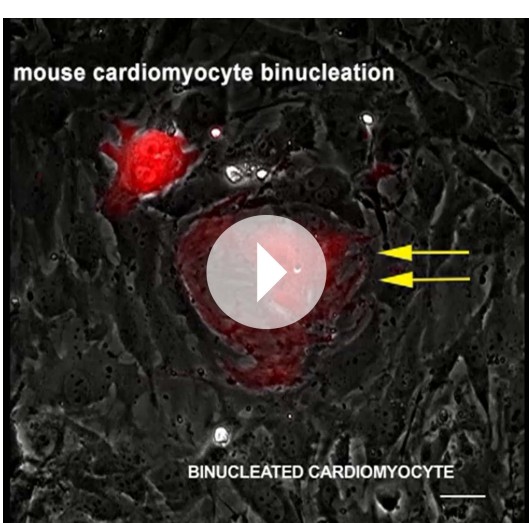

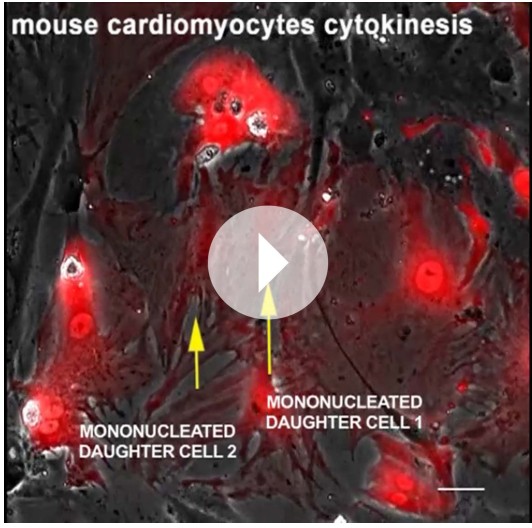

**Video 1.** Mouse cardiomyocyte binucleation. A 24 hr time-lapse video of a representative P1 *Myh6-Cre;R26R-dTomato-lox* CM undergoing karyokinesis followed by binucleation. Time-lapse-10 min. Scale bar: 30 μm.

**Video 2.** Mouse cardiomyocyte cytokinesis. A 48 hr time-lapse video of a representative P1 *Myh6-Cre;R26R-dTomato-lox* CM undergoing karyokinesis followed by cytokinesis. Time-lapse-10 min. Scale bar: 30 μm. One of the mononucleated daughter cells undergoes consecutive cell division, and karyokinesis followed by cytokinesis.

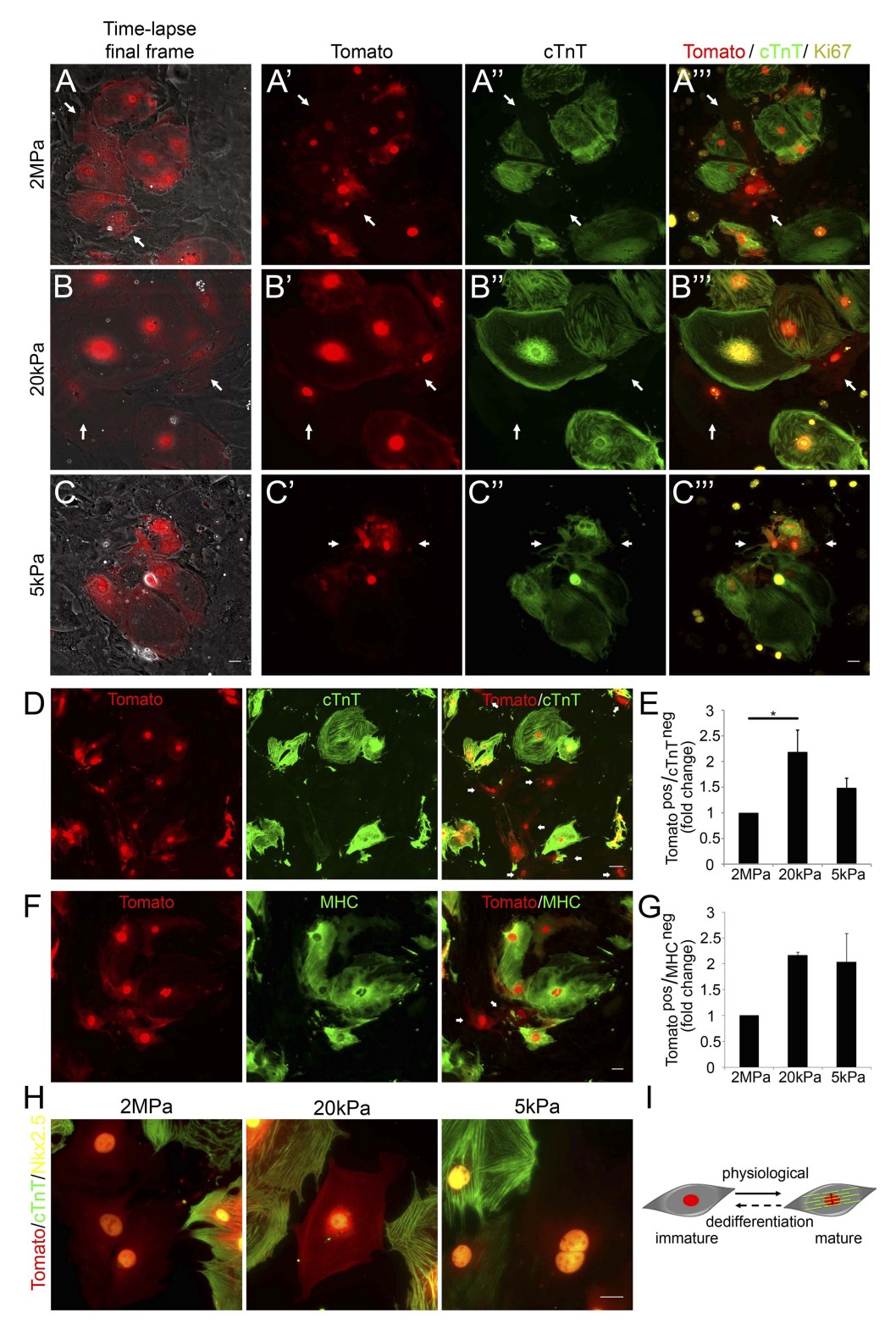

**Figure 3**. Compliant matrices induce cardiomyocyte dedifferentiation. (**A–C**) Correlative live-cell-immunofluorescence of CM dedifferentiation on (**A**) 2 MPa, (**B**) 20 kPa, and (**C**) 5 kPa matrices. Recently divided P1 *Myh6-Cre;R26R-dTomato-lox* CMs, following time-lapse imaging (**A**, **B**, **C**, left panel, and **A'**, **B'**, **C'**), correlated with the expression of cTnT (**A''**, **B''**, **C''**), and Ki67 (**A'''**, **B'''**, **C'''**). CMs following cell division are denoted with white arrows. (**D**) An overview of a field of P1 *Myh6-Cre;R26R-dTomato-lox* CMs (red, left panel), immunostained for cTnT (green, middle panel), and a merge of

*Figure 3. continued on next page*

*Figure 3. Continued*

Tomato+/cTnT− images (right panel). Scale bar: 50 µm. (**E**) Quantification of Tomato+/cTnT− CMs on 2 MPa, 20 kPa, and 5 kPa (n = 19,869). (**F**) An overview of a field of P1 *Myh6-Cre;R26R-dTomato-lox* CMs (red, left panel), immunostained for MHC (green, middle panel), and a merge of the Tomato+/MHC− images (right panel). Scale bar: 20 µm. (**G**) Quantification of Tomato+/MHC− CMs on 2 MPa, 20 kPa, and 5 kPa substrates. Scale bar: 20 µm (n = 13,474). (**H**) P1 *Myh6-Cre;R26R-dTomato-lox*/cTnT− CMs expressing Nkx2.5. (**I**) Schematic diagram showing CM dedifferentiation from a mature to an immature proliferative state, facilitated by compliant matrices. Statistical significance was determined using ANOVA followed by post-hoc Tukey's (HSD) test. Results are marked with one asterisk (*) if p < 0.05.

The following figure supplement is available for figure 3:

**Figure supplement 1**. dTomato cardiomyocytes express cTnT.

---

We next analyzed CM proliferation in the presence or absence of blebbistatin, by quantifying the percentage of Ki67+/cTnT+ CMs (*Figure 4C*). This analysis indicated that blebbistatin-treated CMs display higher cell cycle activity compared to control (untreated) CMs on the 2 MPa and 20 kPa rigidities (*Figure 4C*). Hence, we conclude that relaxation of the acto-myosin myoskelton in CMs by blebbistatin promotes cell cycle re-entry. The lack of blebbistatin effect on the 5 kPa substrate is consistent with the relaxed CM morphology induced by the 5 kPa substrate in the absence of blebbistatin. Taken together, these results suggest that complaint matrices promote CM proliferation primarily via their effect on myoskeletal organization.

## A compliant (20 kPa) matrix promotes clonal expansion of mature cardiomyocytes

A closer examination of CMs derived from the Myh6 lineage that were grown on the 20 kPa matrix and lost cTnT and/or MHC expression, revealed clusters of cells (more than 2), suggesting that these cells were derived from a common CM that underwent multiple cell divisions (*Figure 5—figure supplement 1*). To explore this notion further, we took advantage of the *R26R-Confetti* reporter line (*Snippert et al., 2010*).

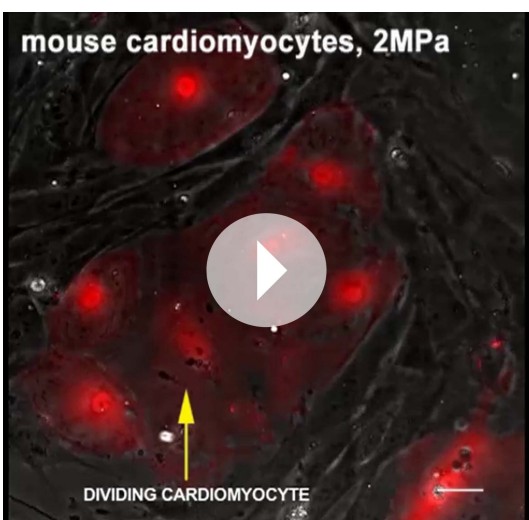 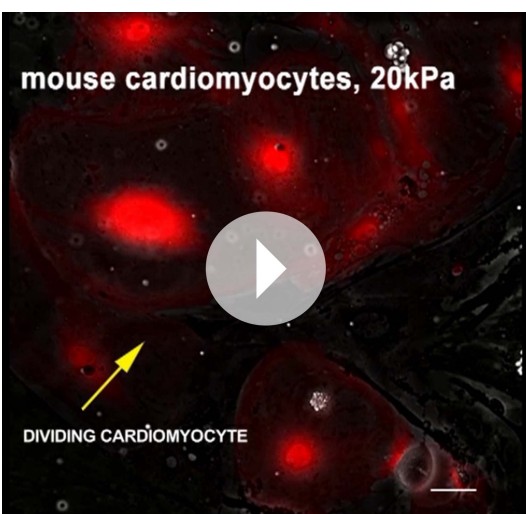

**Video 3.** Mononucleated cardiomyocyte cytokinesis on a 2 MPa substrate. 48 hr time-lapse imaging (with 10 min intervals) of P1 *Myh6-Cre;R26R-dTomato-lox* CMs on a 2 MPa substrate. Scale bar: 30 µm. The last frame of this video is shown in *Figure 3A*, and was correlated with immunofluorescence staining, as shown in *Figure 3A″,A‴*.

**Video 4.** Mononucleated cardiomyocyte cytokinesis on a 20 kPa substrate. 48 hr time-lapse imaging (with 10 min intervals) of P1 *Myh6-Cre;R26R-dTomato-lox* CMs on a 20 kPa substrate. Scale bar: 30 µm. The last frame of this video is shown in *Figure 3B*, and was correlated with immunofluorescence staining, as shown in *Figure 3B″,B‴*.

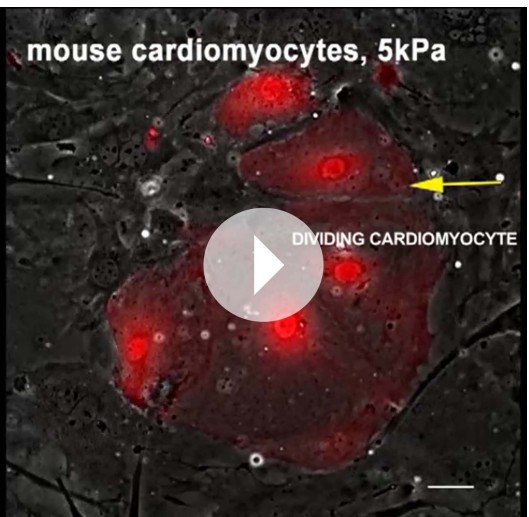

**Video 5.** Mononucleated cardiomyocyte cytokinesis on a 5 kPa substrate. 48 hr time-lapse imaging (with 10 min intervals) of P1 *Myh6-Cre;R26R-dTomato-lox* CMs on a 5 kPa. Scale bar: 30 µm. The last frame of this video is shown in *Figure 3C*, and was correlated with immunofluorescence staining, as shown in *Figure 3C″,C‴*.

In this setting, the *R26R-Confetti* reporter generated stochastic, multicolor Myh6 lineage-derived CMs with four fluorescent proteins, each marking individual clones (*Figure 5A–F*; high magnification areas showing heterogeneous multicolor CM clones in vivo D, E, or individual YFP+ CM clones in F). To asses the clonal potential of CMs grown on the different rigidities, we cultured P1 CMs derived from *Myh6-Cre;R26R-Confetti-lox* hearts, and imaged them at 4, 5, 7, 8 9, and 11 days in culture. At each time point, the exact same fields were imaged, in order to enable the tracing of CM cell division and clonal expansion. Unicolor clones, likely originating from a single CM, were identified on all three rigidities.

We quantified the total amount of CMs that were formed between day 4 and day 11, the percentage of single-color clones, and the number of cells in each clone, for each of the three rigidities (*Figure 5G–I*, respectively). *Figure 5—figure supplement 2*). We found ~2.4-fold higher numbers of total CMs formed on the 20 kPa relative to the 2 MPa substrate (*Figure 5G*). A ~twofold increase in the percentage of unicolor clones was observed on the 20 kPa substrate, compared to both the rigid 2 MPa and softest 5 kPa substrates (*Figure 5H*).

Next, we quantified the number of cells in all the clones, at each time point, and found that the 20 kPa matrix induced an impressive increase in CM cell number (*Figure 5I*). A typical clonal expansion of an individual YFP-positive clone, from only 2 cells at Day 4, to 9 cells within a week (Day 11), is shown (*Figure 5J*). This analysis revealed that neonatal CMs can undergo clonal expansion in vitro, a process which involves multiple cell divisions; moreover, this potential can be induced most readily on the 20 kPa matrix, compared to either rigid or softer matrices.

## Discussion

In this study, we explored the effect of the microenvironment, specifically its rigidity, on CM cell fate, with an emphasis on CM proliferation and dedifferentiation. Notably, most studies that address the impact of rigidity on cell fate decisions have focused on the differentiation processes (*Engler et al., 2008*; *Jacot et al., 2008*; *Bajaj et al., 2010*; *Bhana et al., 2010*). The terminology and characteristics of the reverse process are much less clear, but generally refer to the shift from a differentiated to a less-differentiated cellular stage within the same lineage (*Jopling et al., 2011*). Here, we show that neonatal CMs cultured on compliant (compared to rigid) 2D matrices lose several key manifestations commonly associated with postnatal cardiac muscle differentiation in vivo, such as robust, well-aligned myoskeletel organization, expression of differentiation genes, and cell cycle arrest. Hence, we conclude that compliant substrates promote CM dedifferentiation and proliferation (*Figure 6*). In line with these findings, we demonstrated that dedifferentiated CMs were able to undergo clonal expansion, which involves multiple cell divisions, suggesting a shift to a progenitor cell state.

What is the mechanism underlying the matrix rigidity-dependent CM dedifferentiation process? In particular, we wondered whether there is a clear hierarchy between the mechanical properties of the heart during embryonic and postnatal stages. We propose that the rigidification of the heart after birth from 10-20 kPa to 40–55 kPa (*Berry et al., 2006*; *Engler et al., 2008*; *Jacot et al., 2010*) promotes CM terminal differentiation by triggering the formation of a rigid myoskeleton that mechanically interferes with CM cytokinesis, without affecting karyokinesis. Ultimately, this physiological transition leads to an accumulation of binucleated CMs, correlating with the transition from embryonic hyperplastic growth of CMs, to postnatal hypertrophic growth (*Figure 6*).

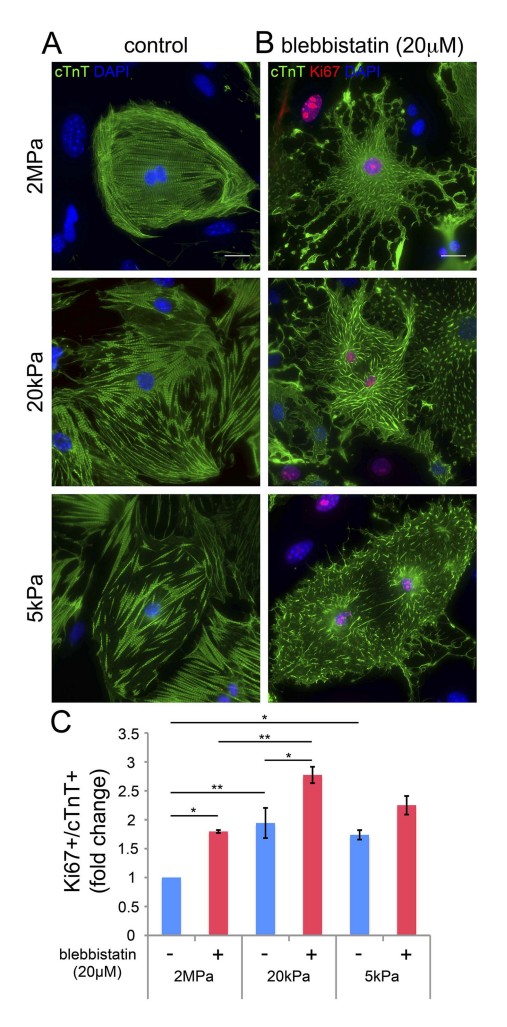

**Figure 4**. Blebbistatin induces CM proliferation. (**A**, **B**) Representative images of P1 mouse CMs cultured in the absence of blebbistatin (**A**), or presence of 20 µM blebbistatin (**B**), on 2 MPa (top), 20 kPa (middle), and 5 kPa (bottom) substrates. Scale bar: 20 µm. (**C**) Quantification of Ki67+/cTnT+ P1 mouse CMs on different rigidities in the presence or absence of 20 µM blebbistatin (n = 25,554). Statistical significance was determined using ANOVA followed by post-hoc Tukey's (HSD) test. Results are marked with one asterisk (*) if $p < 0.05$, and two (**) if $p < 0.01$.

It was previously shown that CM shape is linked to sarcomeric alignment (*Bray et al., 2008*); circular CMs do not assemble actin networks or sarcomeric arrays, whereas rectangular CMs do. Consistent with this, we show high polarity and organization of triangular CMs on rigid substrates, as opposed to low polarity, roundness, and disorganized sarcomeres on compliant substrates. Eukaryotic cell division requires cellular and cytoskeletal remodeling: cells become round and spherical upon entering mitosis (*Heng and Koh, 2010*; *Lancaster et al., 2013*). This mitotic rounding is, to some extent, considered a hallmark of cell division, and the interplay between the actin cytoskeleton, cell shape, and the developing mitotic spindle are all involved in this rounding (*Cadart et al., 2014*; *Lancaster and Baum, 2014*). Thus, it appears that cytoskeletal organization and cell-cycle control are linked together.

In line with this notion, we demonstrate that inhibition of the organized CM contractile apparatus following blebbistatin treatment resulted in increased CM cell cycle re-entry. We show an increase in Ki67-expressing CMs in the presence of blebbistatin on the 2 MPa and 20 kPa rigidities. Blebbistatin treatment strongly perturbed CM myoskeletal organization and consequently induced CM cell cycle activity, suggesting that the development of a robust myoskeleton in CMs is linked to their cell cycle arrest. We suggest that compliant matrices promote CM proliferation primarily via their effect on the organization of the myoskeleton. More broadly, we propose that the rigid matrix increases the forces imposed on the CMs' actin myoskeleton and this facilitates CM differentiation, while inhibition of sarcomeric organization, function and contractility by blebbistatin has the opposite effect, eventually, leading to CM dedifferentiation. That said, it was previously shown that long blebbistatin treatment inhibits cell division, leading to generation of polyploid megakaryocytes (*Shin et al., 2011*). Polyploidy and binucleation are natural outcomes of postnatal CM proliferation events (*Zebrowski and Engel, 2013*). It is possible that the high CM cell-cycle activity observed in the presence of blebbistatin may ultimately result in inhibition of cytokinesis, leading to CM binucleation, or even multinucleation.

One of the most important findings in this study is that cytokinesis, but not karyokinesis, is affected by matrix rigidity: it is facilitated by the compliant substrates, or inhibited by the rigid substrate. Taken together, the data presented here suggest that committed CMs cultured on the 20 kPa (and, to some extent, the 5 kPa) substrate lose the environmental mechanical barrier, leading to the disorganization of the myoskeleton, which facilitates the acquisition of a rounded morphology compatible with cytokinesis and cell division (*Figure 6*).

The mechanisms underlying CM binucleation, due to failure to complete cytokinesis, are poorly understood. Recently, the view that binucleated CMs cannot divide was challenged, showing that

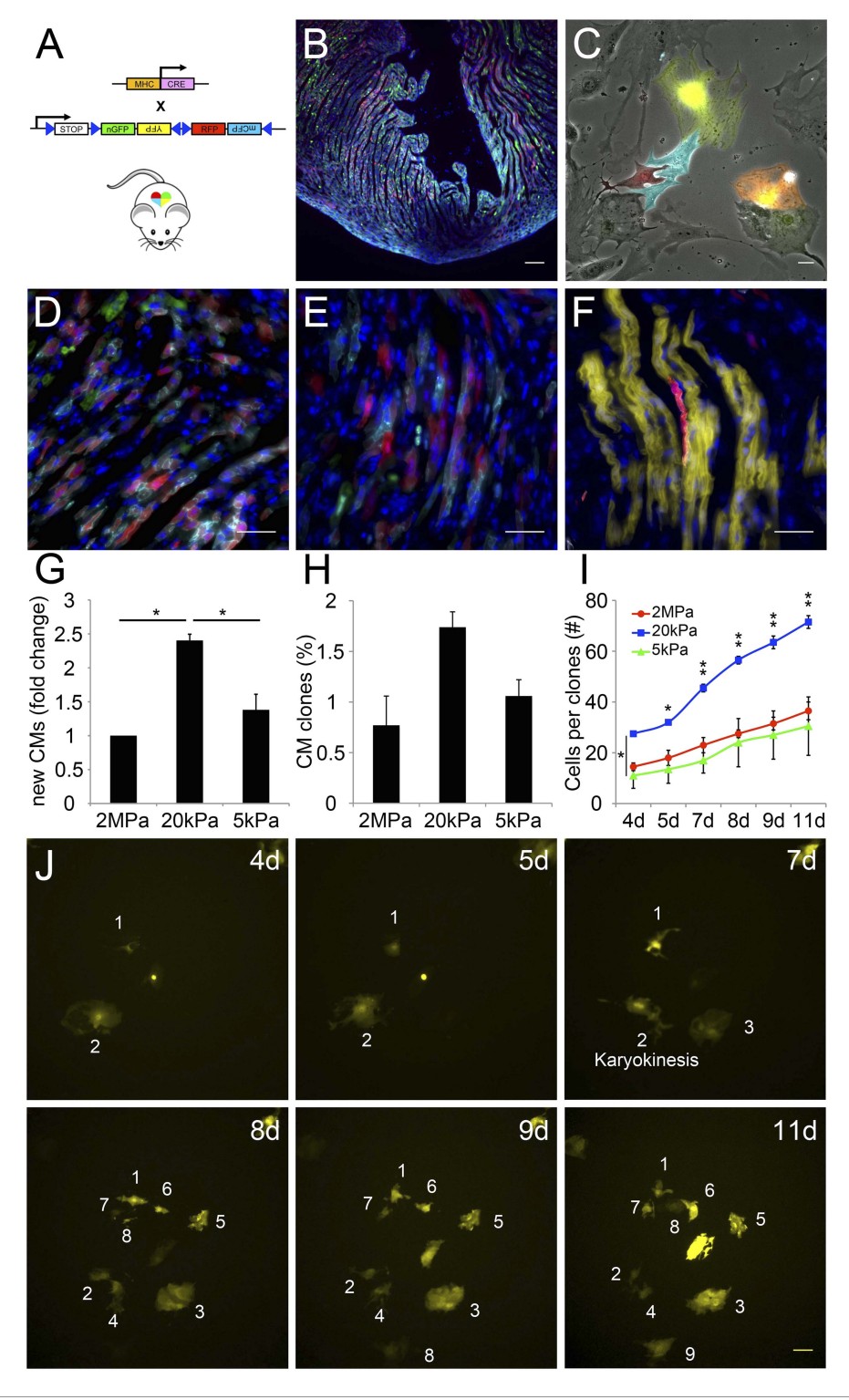

**Figure 5**. The compliant 20 kPa matrix promotes cardiomyocyte clonal expansion. (**A**) A schematic drawing of the clonal tracing strategy of *Myh6-Cre;R26R-Confetti-lox* mice. (**B**) In vivo section of a neonatal *Myh6-Cre;R26R-Confetti-lox* heart. Scale bar: 100 μm. (**C**) In vitro culture of *Myh6-Cre;R26R-Confetti-lox*—derived CMs. Scale bar: 20 μm. (**D–F**) in vivo images of *Myh6-Cre;R26R-Confetti-lox* heart sections at high magnification. Scale bar: 20 μm. (**D–E**) Heterogeneous population of CMs. (**F**) A clone of YFP+ CMs, surrounding a single RFP+ CM. (**G**) 20 kPa substrate facilitates a 2.4-fold increase in CM number, relative to 2 MPa. (**H**) 20 kPa substrate facilitates an increase in
*Figure 5. continued on next page*

*Figure 5. Continued*

the percentage of clones. (**I**) Total number of cells over time on 2 MPa, 20 kPa, and 5 kPa substrates, showing a greater initial number of cells, as well as a greater final number of cells, on 20 kPa substrates (n = 6,401 for G, n = 5,110 for **H**, **I**,). (**J**) Formation of a YFP-positive clone on a 20 kPa substrate, throughout the experiment. Clonal expansion is indicated by an increase in the number of cells. Statistical significance was determined using ANOVA, followed by post-hoc Tukey's (HSD) test. Results are marked with one asterisk (*) if p < 0.05, and two (**) if p < 0.01.

The following figure supplements are available for figure 5:

**Figure supplement 1**. Formation of dedifferentiated cardiomyocyte clones on a 20 kPa matrix.

**Figure supplement 2**. Clonal behavior of individual P1 *Myh6-Cre;R26R-Confetti-lox*—derived cardiomyocytes on different rigidities.

a proliferative burst on P15, induced by a thyroid hormone surge, led to a 1.4-fold increase in CM number (*Naqvi et al., 2014*). We observed rare events of binucleated CMs attempting to divide, though most remain binucleated. However, the few successful division events were facilitated by growth on the 20 kPa matrix, further supporting the notion that CM can respond to matrix rigidity. Interestingly, transgenic expression of a constitutively active (ca) form of ErbB2 in mouse CMs during the neonatal and adult periods could stimulate CM division, expanding the postnatal proliferative and regenerative windows into adulthood. Stimulation of CM division by caErbB2 involved both CM dedifferentiation, and hypertrophy leading to cardiomegaly. In line with the observed dedifferentiation phenotype, caErbB2 signaling promoted proliferation of mono- and bi-nucleated CMs (*D'Uva et al., 2015*).

In other model systems, rigidity was shown to influence proliferation and stemness of cells. In line with our data, mouse embryonic fibroblasts (MEFs), vascular smooth muscle cells (VSMCs), and MCF10A cells proliferate best on ~24 kPa (*Klein et al., 2009*). Moreover, muscle stem cells (MuSCs)

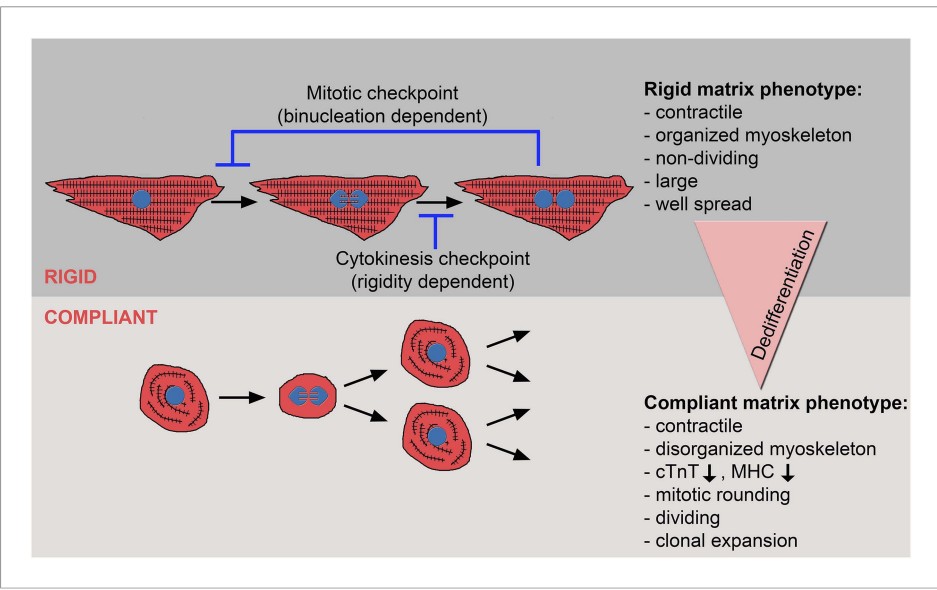

**Figure 6**. The crosstalk between matrix rigidity and CM cell fate. A schematic model showing the effect of matrix rigidity on CM cell fate. Rigid substrates maintain the differentiated state of CMs, which is incompatible with cytokinesis. Blocked cytokinesis leads to binucleated CMs. A mitotic checkpoint determines the ability of CMs to re-enter the cell cycle. Compliant substrates control a second checkpoint, mechanical in nature, which enables cytokinesis. Compliant substrates induce CM dedifferentiation into a less organized, immature and proliferative state. This state involves cytoskeletal disarrangements leading to CM mitotic rounding, thus enabling cell division.

that were cultured on a soft substrate that mimics muscle tissue rigidity (~12 kPa) display delayed differentiation, self-renewal, and maintained stemness (*Gilbert et al., 2010*). In agreement with our findings, these researchers propose that decreased rigidity preserves stemness by altering cell shape, resulting in cytoskeletal rearrangements.

In both neonatal mice and adult zebrafish, cardiac regeneration processes were accompanied by loss of sarcomeric structures and re-acquisition of cell division, both attributed to CM de-differentiation (*Poss, 2007*; *Jopling et al., 2010*; *Porrello et al., 2011*). Our correlative live-imaging and immunostaining results support the notion that dedifferentiated CMs disassemble their sarcomeres, in order to divide.

The fact that CMs have lost MHC or cTnT expression, and the partial disassembly of the sarcomeres during cytokinesis, further support our model of CM dedifferentiation, rather than the presence of progenitor cells. The *Myh6-Cre;R26R-confetti-lox* system enabled us to document clonal behavior in neonatal CMs. We show that clonal expansion is rigidity-dependent, with an optimal clonal efficiency on the 20 kPa. Clone formation originated at various time points and progressed at different rates, suggesting a continuous proliferative capacity, rather than a final 'burst' of proliferation. Whether neonatal or adult CMs are capable of undergoing multiple cycles of cell division in vivo, has yet to be determined.

Our findings bear obvious relevance for cardiac regeneration after injury. In this unfortunate event, CMs in the vicinity of the scar tissue sense a stiffer environment, compared to the healthy adult heart (*Jacot et al., 2010*). Thus, if fully differentiated CMs present in the vicinity of the damaged area, can still maintain the capacity to dedifferentiate and resume cell division if confronted with a sufficiently compliant scaffold, a new avenue toward the regeneration of heart tissue may be opened.

## Materials and methods

### Isolation and culture of neonatal rat cardiomyocytes

Experiments were approved by the Animal Care and Use Committee of the Weizmann Institute of Science. Neonatal rat CMs were isolated and cells were cultured, as previously described (*Shneyvays et al., 2002*). Briefly, 1-day-old newborn Wistar rats were decapitated; the hearts were harvested, cut into small pieces (1–2 mm), and washed several times in phosphate-buffered saline (PBS) to remove excess blood. To obtain a CM-rich culture, the hearts were digested in RDB, a proteolytic enzyme extracted from the fig tree (Biological Institute, Nes Ziona, Israel) in 6–8 cycles of 10 min each, at room temperature. The cells were centrifuged at 1500 rpm for 5 min. The pellet was resuspended and cultured in a 10 cm dish for 30 min-1 hr for pre-plating, to produce a CM-rich culture. CMs were then cultured on glass or PDMS surfaces. CMs began to beat spontaneously within 2–3 days, on all of the tested substrates.

### Isolation and culture of neonatal mouse cardiomyocytes

CM isolation was performed using the Neonatal Heart Dissociation Kit (gentleMACS, Miltenyi Biotec, Auburn, CA, USA). CMs were cultured in DMEM/F12 containing Na-pyruvate, non-essential amino acids, penicillin (100 U/ml), streptomycin (100 µg/ml), 2 mM L-glutamine, 5% horse serum, and 20% fetal bovine serum (FBS). Briefly, 1-day-old newborn wild-type (WT) or *Myh6-Cre;R26R-tdTomato-lox* CMs were decapitated; the hearts were harvested and transferred into a 10 cm dish containing PBS, and remaining blood was carefully pumped. Hearts were transferred into the gentleMACS C Tube, and enzyme mix was added. C Tubes containing hearts and enzyme mix were incubated for 15 min at 37 ˚C, transferred to the gentleMACS Dissociator, and the gentleMACS Program 'mr_neoheart_01' was run. These incubation and dissociation steps were repeated 2–3 times. C tubes containing hearts were centrifuged at 2000 rpm for 5 min. The pellet was resuspended and cultured in a 10 cm dish for 30 min-1 hr for pre-plating, to produce a CM-rich culture. CMs were then cultured on fibronectin-coated PDMS surfaces. CMs began to beat spontaneously within 2–3 days, on all of the tested substrates.

### Transgenic animals

For live-cell imaging and imuunofluorescence assays of Myh6-expressing cells, we crossed mice carrying the Cre coding sequence inserted after the alpha myosin heavy chain promoter (Myh6-cre), which can drive high-efficiency gene recombination in CMs (*Agah et al., 1997*), with ROSA26-flox-STOP-tdTomato indicator mice (R26R-tdTomato) (*Madisen et al., 2010*), thus creating red fluorescent-labeled CMs that

could be tracked. For clonal analysis of Myh6-expressing cells, Myh6-Cre mice were crossed with the ROSA26-flox-STOP-Confetti reporter mice (*Snippert et al., 2010*). ROSA26-flox-STOP-Confetti reporter mice (R26R-confetti reporter mice) were kindly provided by Dr. Shalev Itzkovitz, Weizmann Institute of Science.

## Clonal analysis of Myh6-expressing cells

*Myh6-Cre;R26R-confetti-lox* CMs were cultured on different PDMS matrices on grid-bottomed MatTek dishes. Using a DeltaVision Elite system (Applied Precision, USA) on an Olympus IX71 inverted microscope, running softWoRx 6.0, fluorescence images were acquired at 10× or 20× magnifications, by a CoolSnap HQ2 CCD camera (Roper Scientific, USA). For clonal analysis, we utilized the DeltaVision collect panels option. Panel stitching was performed using softWoRx.

## Preparation of substrates of varying rigidities

Polydimethylsiloxane (PDMS) substrates of varying rigidities were prepared, using a Sylgard 184 silicone elastomer kit (Dow Corning, USA). The silicone elastomer component was mixed with the curing agent, degassed, and spin-coated at 2000 rpm for 2 min with a Spin Processor WS-650MZ-23NPP/LITE (Laurell Technologies) on MatTek #0, #1 or MatTek #1.5 glass-bottomed dishes (MatTek Corporation), or microscopy coverslips (Electron Microscopy Science) for live-cell and for immunofluorescence experiments, or MatTek gridded #2 glass-bottomed dishes for the correlation live-cell immunofluorescence experiments, to obtain a $35 \pm 5$ μm-thick PDMS layer. Subsequently, crosslinking of the elastomer was carried out at 70°C overnight. The compliance of the PDMS substrates was verified by using the methodology as previously described (*Prager-Khoutorsky et al., 2011*). Briefly, polymerized slabs of PDMS were used for bulk measurements of Young's moduli, using an Instron universal testing machine (Instron). Elastomer to curing agent ratios of 10:1, 50:1 and 75:1 corresponded to Young's moduli of 2 MPa, 20 kPa and 5 kPa, respectively. Dishes with a layer of PDMS were functionalized with 20 μg/ml fibronectin at 4°C overnight. Before cell plating, plates were washed with PBS and growth medium.

## Drug treatment

CMs were cultured for 72 hr, and then treated with the myosin II inhibitor blebbistatin (20 μM, Sigma) for 24 hr. Following blebbistatin treatment, cells were permeabilized, fixed, and immunostained for sarcomere and proliferation markers.

## Immunocytochemistry

Cells were permeabilized with 3% paraformaldehyde (PFA) in PBS containing 0.25% Triton X-100 for 3 min, and then fixed with 3% PFA in PBS for 20–30 min, washed 3 times in PBS, blocked with 10% goat serum in PBS, and stained for the following sarcomeric markers: cardiac troponin T (cTnT, abcam); MF20 (MHC, Myosin heavy chain Developmental Studies Hybridoma Bank [DSHB]); cardiac MHC (Abcam); myomesin (DSHB); Nkx2.5 (Abcam), and the following proliferation markers: Ki67 (Abcam) and phospho-histone-3 (PH3, Santa Cruz). Secondary antibodies used were Alexa-488, Cy3, Alexa-647, conjugated anti-mouse IgG1, and Cy5-conjugated anti-mouse IgG2b. Cells were also stained for DAPI, to visualize the nuclei.

## Fluorescence microscopy and live-cell imaging

Live-cell imaging and sample examination were performed using a DeltaVision Elite system (Applied Precision, USA), on an Olympus IX71 inverted microscope, running softWoRx 6.0. Fluorescent images were acquired at 10×, 20×, 40× and 60× magnifications, by a CoolSnap HQ2 CCD camera (Roper Scientific, USA). Time-lapse imaging was carried out for 24 hr or 48 hr at 10 min intervals, and acquired at a 20× or 10× magnification (20×/0.5NA or 10×/0.3NA objectives). Correlated live-cell immunofluorescence microscopy was performed on gridded #2 glass-bottomed dishes (MatTek Corporation), using either 10×/0.3NA or the 20×/0.5NA objectives for time-lapse imaging, and the 10×/0.3NA or 20×/0.85NA objectives for immunofluorescence correlation.

## Image analysis

Images of *Myh6-Cre;R26R-confetti-lox* mice in vivo sections were taken with a Nikon Ti-E inverted fluorescence microscope equipped with a 10×, or 100× oil-immersion objectives and a Photometrics

Pixis 1024 CCD camera using MetaMorph software (Molecular Devices, Downington, PA). The image-plane pixel dimension was 1.3 µm for 10× magnification, and 1.3 µm for 100× magnification. CM projected area and best-fit ellipse aspect ratio of cardiac troponin T-stained cells were calculated, using ImageJ software.

### Video editing

Descriptions, titles and arrows have been added to all videos using 'Final Cut Pro' editing software.

### Statistical analysis

Generally, all experiments were carried out with $n \geq 3$. In all panels, numerical data are presented as mean +s.e.m. Statistical significance was determined using ANOVA, followed by post-hoc Tukey's (HSD) test. Results are marked with one asterisk (*) if $p < 0.05$, two (**) if $p < 0.01$, and three (***) if $p < 0.001$.

## Acknowledgements

This work was supported by grants to ET from the European Research Council (ERC Grant 281289, CMturnover) and the Israel Science Foundation, and FP7 grants to BG: EU 710639 (NanoCARD), an ERC project (294852, SynAd), and the Israel Science Foundation. We thank Dr. Shalev Itzkovitz for his help in providing the R26R-confetti reporter mice, and Ayal Ronen and Barbara Morgenstern for expert editorial assistance. We also thank Omri Yahalom, who assisted with the statistical methodology. BG holds the Erwin Neter Chair in Cell and Tumor Biology.

## Additional information

### Funding

| Funder | Grant reference | Author |
|---|---|---|
| European Research Council (ERC) | ERC Grant 281289, CMturnover | Eldad Tzahor |
| Israel Science Foundation (ISF) | ISF | Eldad Tzahor, Benjamin Geiger |
| European Commission (EC) | EU 710639 (NanoCARD) | Benjamin Geiger |
| European Research Council (ERC) | ERC Grant 294852, SynAd | Benjamin Geiger |

The funders had no role in study design, data collection and interpretation, or the decision to submit the work for publication.

### Author contributions

YY-R, Conception and design, Acquisition of data, Analysis and interpretation of data, Drafting or revising the article; DR, Acquisition of data, Analysis and interpretation of data; RS, Analysis and interpretation of data, Drafting or revising the article; BG, ET, Conception and design, Analysis and interpretation of data, Drafting or revising the article

### Ethics

Animal experimentation: The experiments were approved by the Animal Care and Use Committee of the Weizmann Institute of Science.

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
