## [Decision Letter]

Thank you for submitting your work entitled “Reduced matrix rigidity promotes cardiomyocyte dedifferentiation, proliferation and clonal expansion” for peer review at *eLife*. Your submission has been favorably evaluated by Fiona Watt (Senior Editor), a Reviewing Editor, and three reviewers.

The reviewers have discussed the reviews with one another, and the Reviewing editor has drafted this decision to help you prepare a revised submission

The reviewers were generally favorable about the importance of the study and the conclusions. However, there were several issues that tempered enthusiasm and aspects of the data that did not fully support the conclusions. In particular, issues around the cell types, the rigor with which quantitative analysis is performed, and over interpretation of data need to be addressed. Essential revisions are below:

1) In order to definitively assess CM proliferation, the authors need to demonstrate increase in CM numbers in addition to staining for markers.

2) Based on the limited data presented, it is quite an overly strong statement to say that cell rounding is a “prerequisite” for induction of CM cytokinesis. Similarly in the Discussion, it is too strong to say that cytoskeleton and cell cycle are “inseparable” based on the experiments performed.

3) It doesn't appear that the authors measured the compliance of the materials they performed the studies with. This needs to be done to ensure the accuracy of the effects of the reported stiffness (modulus) values. It is not sufficient to rely on previously reported values, and no citation is provided along with the mixture ratios and values provided.

4) Quantitative methods are generally weak and statistical analyses were not performed correctly. *T*-tests can only be used to compare between two experimental groups. In the case of three or more groups, ANOVA followed by appropriate post-hoc tests must be performed to determine statistical significance. In several cases, no indication of statistical significance is denoted in the figures, yet the authors still make statements comparing results based on absolute values. The authors fail to denote the “n” number of cells throughout most of the manuscript for their various experiments. This critical detail should be provided consistently throughout to ensure all of the results are based on sufficient numbers of cell events. Some of the results appear to be based on a very small number of cells – Figure 2 has fewer than ten events for each group.

5) An alternative mechanism for more proliferation on intermediate softness matrices is that cells are more upright and rounded (like in division) rather than spread as on rigid substrate, but in this case the cell shape on 20kPa needs quantitation (add to Figure 1). In addition, cell division should be studied on micropatterned substrates.

6) Since the highly proliferative cardiac fibroblasts and the “non-dividing” cardiomyocytes might share a common progenitor, the authors should prove the specificity of their αMHC driven expression of tdTomato (or is the expression leaky?). In other words, do all tdTomato cells stain for cardiac-specific MHC and/or Troponin? In this respect, Figure 3, showing more tdTom cells than with cardio markers, should be explained better.

7) The finding that “rigid matrix promotes karyokinesis without cytokinesis, leading to CM binucleation (Figure 2)” (in the subsection “Distinct rigidity-dependent cardiomyocyte proliferation mechanisms”) seems consistent with results for megakaryocytes cultured on stiff rather than soft gels with high collagen (40), but this latter paper used the myosin-II inhibitor blebbistatin as well as collagen density (for cell spreading control) to elucidate some aspects of mechanism. The observations in the present submission by Yahalom-Ronen et al. could benefit from similar (or other) approaches to clarifying some aspect of mechanism.

---

## [Author Response]

The reviewers were generally favorable about the importance of the study and the conclusions. However, there were several issues that tempered enthusiasm and aspects of the data that did not fully support the conclusions. In particular, issues around the cell types, the rigor with which quantitative analysis is performed, and over interpretation of data need to be addressed. Essential revisions are below.

We sincerely thank the reviewers for their thoughtful comments and suggestions. In the current version of the manuscript, we revised the statistical analyses. Specifically, we conducted ANOVA followed by definitive Post-Hoc tests, both of which yield significant P-values. Importantly, we added new experimental data, addressing: 1) the number of cardiomyocytes (CMs); 2) the effect of actomyosin relaxation (by adding blebbistatin) on CM myoskeleton architecture and proliferation; 3) the influence of the compliant 20kPa substrate on the behavior of the rat neonatal CMs; and 4) the lack of “leakiness” of αMHC-Cre-driven tdTomato fluorescent protein expression in CMs. Finally, we revisited some of our earlier statements, as suggested by the reviewers, and modified them, where necessary.

*1) In order to definitively assess CM proliferation, the authors need to demonstrate increase in CM numbers in addition to staining for markers*.

To directly address the reviewers’ comment, we performed a large-scale live-cell imaging analysis of CMs cultured on PDMS displaying the various rigidities. We then traced and quantified CM cytokinesis events that resulted in two daughter CMs. To do so, we performed 6 independent time-lapse experiments. For each experiment, we cultured CMs derived from P1 transgenic *αMHC-Cre;tdTomato-lox* mice on 2 MPa, 20kPa, and 5kPa. After 4 days, we performed time-lapse imaging of the CMs for 48 hours, with 10 min intervals between images. For all rigidities in each experiment, we imaged between 15-30 fields, ranging from 300 to over 700 cells, for a total of 2,878 CMs. In these experiments we counted the number of new CMs and calculated their percentage formed on the two compliant surfaces compared to cells growing on the rigid 2MPa substrate. The results of this analysis appear in new Figure 2.

To further quantify the actual increase in CM number, we used the *R26R-Confetti* reporter line, and counted *αMHC-Cre;Confetti-lox* -derived CMs growing on 2MPa, 20kPa, and 5kPa surfaces, between days 4 -11 in culture (over 5,000 CMs). Part of the analysis of the clonal expansion was included in the original submission (Figure 4). In light of reviewer’s comment regarding total CM numbers, formed on the different surfaces, we have now calculated the addition of CMs between days 4 and 7 following cell seeding. This new analysis indicates a significant increase in CM number on the 20kPa and 5kPa substrates (2.4- and ∼1.4-fold, respectively), compared to the “rigid” 2MPa substrate (Figure 4 new panel). These values are in agreement with the increase in proliferation markers, counting of cytokinesis events, and clonal expansion of CMs cultured on the soft matrices.

*2) Based on the limited data presented, it is quite an overly strong statement to say that cell rounding is a “prerequisite” for induction of CM cytokinesis. Similarly in the Discussion, it is too strong to say that cytoskeleton and cell cycle are “inseparable” based on the experiments performed*.

We rephrased the relevant sections in the Results and Discussion sections, indicating that our findings are in line with the view that cell rounding and the decrease in myoskeleton organization correlate well with increased cytokinesis, and refrain from strong causal claims that go beyond the currently available evidence.

*3) It doesn't appear that the authors measured the compliance of the materials they performed the studies with. This needs to be done to ensure the accuracy of the effects of the reported stiffness (modulus) values. It is not sufficient to rely on previously reported values, and no citation is provided along with the mixture ratios and values provided*.

The compliance of the PDMS substrates was verified, using a methodology previously employed by [38] conducted in our (the Geiger) lab. Briefly, polymerized slabs of PDMS were subjected to bulk measurements of Young’s moduli, using an Instron Universal Testing System. The elastomers prepared for this study had curing agent ratios of 1:10, 1:50 and 1:75, corresponding to Young’s moduli of 2MPa, 20kPa and 5kPa, respectively. Specific batches of Sylgard^®^ 184 (Dow Corning), including those used in this study, are routinely tested in our lab. This information was added to the Methods section.

*4) Quantitative methods are generally weak and statistical analyses were not performed correctly.* T*-tests can only be used to compare between two experimental groups. In the case of three or more groups, ANOVA followed by appropriate post-hoc tests must be performed to determine statistical significance. In several cases, no indication of statistical significance is denoted in the figures, yet the authors still make statements comparing results based on absolute values. The authors fail to denote the “n” number of cells throughout most of the manuscript for their various experiments. This critical detail should be provided consistently throughout to ensure all of the results are based on sufficient numbers of cell events. Some of the results appear to be based on a very small number of cells –*
Figure 2
*has fewer than ten events for each group*.

In the revised manuscript, statistical significance was determined using ANOVA followed by post-hoc Tukey’s (HSD) tests. We further improved the overall validity of the analyzed data by increasing the number of biological repeats in several experiments, and providing the statistical parameters both within the figures themselves, and in the relevant text. We also added the number of cells examined in each experiment throughout the manuscript. Notably, our findings were based on hundreds to thousands of CMs analyzed per experiment. The quantifications of karyokinesis, cytokinesis, and binucleation events described in Figure 2 were obtained from 5 independent time-lapse experiments, and based on the tracking of over 2,000 CMs by analyzing thousands of movies.

It is noteworthy that some events, such as definitive cytokinesis of mature, binucleated CMs, were quite rare events, and observed in only a few cases in our movies. The fact that binucleated CMs can undergo cytokinesis and divide is a highly unexpected event, too rare to be subjected to a standard significance test. In these unusual cases, we refer in the text to the number of events, without calculating their statistical significance. We highlight and further discuss these considerations in the revised manuscript. These results are now moved as a new panel in Figure 2—figure supplement 1.

*5) An alternative mechanism for more proliferation on intermediate softness matrices is that cells are more upright and rounded (like in division) rather than spread as on rigid substrate, but in this case the cell shape on 20kPa needs quantitation (add to*
Figure 1*). In addition, cell division should be studied on micropatterned substrates*.

As requested, we performed additional experiments with compliant substrates on rat neonatal CMs. In Figure 1, we now provide further quantifications and statistics of CM axial ratio (“polarization”) and projected area (“spreading”) on 2MPa, 20kPa, and 5kPa substrates. We found that CMs cultured on 20kPa are less polarized than those on 2MPa, but more polarized than those on 5kPa substrates. We observe that CMs on the 20kPa are less spread and more rounded, compared to those cultured on the rigid substrate. However, CMs growing on the 5kPa substrate, although smaller and more rounded than the CMs growing on the 20kPa surfaces, were less proliferative than the CMs cultured on the 20kPa substrate. We cannot yet definitively explain the apparent differences between the rounding and cytoskeletal effects induced by the 5kPa and the 20kPa surfaces, and the effects of the two surfaces on cytokinesis. It is possible that the 5kPa substrate is simply “too soft” for CMs, and cannot support the formation of the actomyosin-based contractile ring, essential for successful cytokinesis. We refer to this issue in the revised text.

The reviewers’ suggestion to study cell division on micropatterned substrates is interesting, but we think that it would exceed, by far, the current scope of this research, which focuses on the effects of matrix rigidity on CM cell fates.

*6) Since the highly proliferative cardiac fibroblasts and the “non-dividing” cardiomyocytes might share a common progenitor, the authors should prove the specificity of their αMHC driven expression of tdTomato (or is the expression leaky?). In other words, do all tdTomato cells stain for cardiac-specific MHC and/or Troponin? In this respect,*
Figure 3*, showing more tdTom cells than with cardio markers, should be explained better*.

The αMHC-Cre-driven expression of tdTomato is very commonly used in the cardiac regeneration field to label the CM lineage. That said, we now provide further evidence that αMHC-Cre-driven tdTomato expression is not leaky, and is restricted to CMs.

First, we co-stained αMHC-Cre dTomato cells with anti-cTnT antibody. The staining showed an overlap of 99.62% of dTomato^+^/cTnT^+^ CMs, with less than 1% of the dTomato CMs negative for cTnT (Figure 3—figure supplement 1). Secondly, Nkx2.5 is a well-known cardiac marker that identifies embryonic and mature CMs, as well as cardiac progenitors. We therefore co-stained dTomato CMs with Nkx2.5, and found that all dTomato CMs were also positive for Nkx2.5 (Figure 3). Taken together, these results strongly demonstrate the “clean” genetic lineage marker system that we used for labeling CMs.

The definition of CM “terminal differentiation” is quite elusive (49). CM differentiation is often characterized by binucleation, hypertrophic growth potential, withdrawal from the cell cycle, and increased myofibrillar organization (49). Dedifferentiation is often defined as a functional and morphological regression of a mature cell within its own lineage (21). Moreover, in both zebrafish and mouse models, dedifferentiation was defined by a partial and transient disassembly of the sarcomeric structure (22; 35).

As explained above, “dTomato” cells represent CMs that express, or have previously expressed, the αMHC gene, a signature of CM differentiation. Under regular conditions, we detected almost 100% of dTomato^+^/cTnT+ CMs. Therefore, dTomato-positive CMs, which were cTnT^-^ or MHC^-^ (protein), reflect the loss of sarcomeric markers within the CM lineage, which is consistent with the definition of CM dedifferentiation, among other parameters such as cell cycle re-entry. We clarify our explanations of this type of analysis in the revised manuscript.

*7) The finding that “rigid matrix promotes karyokinesis without cytokinesis, leading to CM binucleation (*Figure 2*)” (in the subsection “Distinct rigidity-dependent cardiomyocyte proliferation mechanisms”) seems consistent with results for megakaryocytes cultured on stiff rather than soft gels with high collagen (*[40]*), but this latter paper used the myosin-II inhibitor blebbistatin as well as collagen density (for cell spreading control) to elucidate some aspects of mechanism. The observations in the present submission by Yahalom-Ronen et al. could benefit from similar (or other) approaches to clarifying some aspect of mechanism*.

We appreciate this suggestion to investigate some aspect of the mechanism by using the myosin-II inhibitor blebbistatin, which was previously shown to inhibit actin-myosin interactions in mouse CMs (12). Specifically, we explored whether disruption of the highly organized architecture of the CM myoskeleton would induce cell cycle reentry. Towards that end, we used the methodology employed by Shin and colleagues (40), by culturing 1-day-old neonatal CMs on substrates of varying rigidities (2MPa, 20kPa, 5kPa), allowing them to adhere to the substrate for 72h, and then treating them with 20µM blebbistatin for 24 hours. We then fixed the cells and immunostained them for cTnT and Ki67. We found that blebbistatin severely disrupts the CM myoskeleton, leading to massive disorganization in the sarcomeric system. Blebbistatin treatment further induced a major shift in CM cell shape, manifested by major flattening, dramatic fragmentation of the sarcomeric arrays, and nearly complete loss of striations (new Figure 4). Taken together, our results indicate that blebbistatin treatment results in severe disorganization of contractile myofilaments in CMs (new Figure 4).

Since our findings strongly suggest that disruption of the CM myoskeleton facilitates, or is linked to, CM cell cycle re-entry and overall CM dedifferentiation, we analyzed CM cell cycle activity in the absence or presence of blebbistatin, by quantifying the percentage of Ki67^+^/cTnT^+^ cells (n=25,544). This analysis indicated that blebbistatin-treated CMs display higher cell cycle activity compared with control (untreated CMs) on the 2MPa and 20kPa rigidities (Figure 4). Hence, we conclude that relaxation of the myoskelton in CMs by blebbistatin promotes cell cycle re-entry. The lack of blebbistatin effect on the 5kPa substrate is consistent with the “already-relaxed” morphology of CMs cultured on the 5kPa substrate.

We suggest that compliant matrices promote CM proliferation primarily via their effect on the mechanics and organization of the myoskeleton. More broadly, we propose that the rigid matrix enhances the forces imposed on the CMs’ actin myoskeleton and this facilitates CM differentiation. In contrast, inhibiting CM sarcomeric organization and function (i.e. contractility) by blebbistatin has the opposite effect of inducing CM dedifferentiation. These results point the way toward additional avenues of investigation, which we plan to pursue in greater detail in a follow-up study.